# Synergy between the small intrinsically disordered protein Hsp12 and trehalose sustain viability after severe desiccation

Skylar Xantus Kim, Gamze Çamdere, Xuchen Hu, Douglas Koshland*, Hugo Tapia*

Department of Molecular and Cell Biology, University of California, Berkeley, Berkeley, United States

**Abstract** Anhydrobiotes are rare microbes, plants and animals that tolerate severe water loss. Understanding the molecular basis for their desiccation tolerance may provide novel insights into stress biology and critical tools for engineering drought-tolerant crops. Using the anhydrobiote, budding yeast, we show that trehalose and Hsp12, a small intrinsically disordered protein (sIDP) of the hydrophilin family, synergize to mitigate completely the inviability caused by the lethal stresses of desiccation. We show that these two molecules help to stabilize the activity and prevent aggregation of model proteins both in vivo and in vitro. We also identify a novel in vitro role for Hsp12 as a membrane remodeler, a protective feature not shared by another yeast hydrophilin, suggesting that sIDPs have distinct biological functions.

DOI: https://doi.org/10.7554/eLife.38337.001

*For correspondence:
koshland@berkeley.edu (DK);
hugo.tapia@berkeley.edu (HT)

Competing interests: The authors declare that no competing interests exist.

## Introduction

In the near future, global food security will be challenged by the effect of climate change on crop yields (*Schwalm et al., 2017*; *Tirado et al., 2013*). Potential solutions to more frequent drought may come from the study of diverse organisms, collectively named anhydrobiotes, which can survive extreme water loss. Anhydrobiotes include single celled organisms (bacteria and yeast), simple multi-cellular organisms (plant seeds), as well as complex multicellular plants and animals (resurrection ferns, tardigrades and nematodes). The acquisition of a desiccation tolerant state of all these organisms requires pre-exposure to a stress condition like nutrient starvation or reduced water that induces dramatic changes in the profile of gene expression and cellular constituents. Among all these changes, the identity of the stress effectors that are necessary and sufficient to mitigate all the stresses of desiccation remained unknown. Furthermore, the removal of water from organisms has the potential to induce many potential stresses, given the ubiquitous function of water in so many biological processes. Which of these potential stresses must be mitigated to allow desiccation tolerance also remains incomplete.

An important clue to the identification of critical stress effectors of desiccation came from the discovery that almost all anhydrobiotes have high levels of the disaccharide trehalose and small, hydrophilic, and intrinsically disordered proteins collectively called hydrophilins (*Crowe et al., 1992*; *Potts, 2001*). Recently, the in vivo requirements for trehalose and hydrophilins in desiccation tolerance have been addressed in nematodes, yeast and tardigrades that lack trehalose or hydrophilins because of either mutations or gene knock-downs using RNAi (*Boothby et al., 2017*; *Erkut et al., 2011*; *Tapia and Koshland, 2014*). The abrogation of these factors caused diverse impacts on desiccation tolerance, as measured by the organism's survival to drying. Nematodes lacking trehalose are extremely sensitive to short term desiccation, while yeast lacking trehalose are sensitive to only long term-tolerance desiccation (*Erkut et al., 2011*; *Tapia and Koshland, 2014*). At least one species of

tardigrade is desiccation tolerant although naturally lacking trehalose (*Boothby et al., 2017*). Loss of individual different hydrophilins only partially reduces tolerance in nematodes, tardigrades, and yeast, and never to the degree as observed when lacking trehalose (*Boothby et al., 2017*; *Calahan et al., 2011*; *Erkut et al., 2013*). Indeed, in nematodes, even when a specific hydrophilin is missing, >10% of nematodes survive, demonstrating moderately high levels of desiccation tolerance never observed in sensitive species (*Erkut et al., 2013*). The fact that loss of trehalose or individual hydrophilins can reduce desiccation tolerance shows they can mitigate one or more stresses of desiccation. However, the variability of their impact suggests that unknown combinations of trehalose, hydrophilins or additional, as yet unidentified, stress effectors must help mitigate the same or different lethal stresses of desiccation.

To complement these loss of function tests, we developed a system in budding yeast to examine the combinatorial roles of stress effectors in desiccation. Only 0.001% of exponentially dividing yeast cells survive desiccation compared to 20–40% survival of stationary phase cells (*Calahan et al., 2011*). To assess the sufficiency of different stress effectors (e.g. of trehalose or hydrophilins) to confer desiccation tolerance, we engineered exponentially dividing cells to have increased levels of only trehalose or individual tardigrade hydrophilins (independent of all the other cellular factors that differ between exponential and stationary phase cells) (*Boothby et al., 2017*). The desiccation tolerance of these cells increased by 1000-fold, but remained one to two orders of magnitude less than the tolerance naturally reached in stationary phase cells (*Boothby et al., 2017*; *Tapia et al., 2015*). These observations further reinforce the notion that robust desiccation tolerance likely requires an unknown combination of trehalose, hydrophilins or other stress effectors working in concert to alleviate one or more stresses imposed by complete water loss.

What are the stresses imposed by desiccation? Given the ubiquitous molecular functions of water in biological processes, its absence has been proposed to induce a plethora of stresses. After drying, desiccation sensitive nematodes and yeast exhibit membrane blebbing and protein aggregation, respectively (*Erkut et al., 2011*; *Tapia and Koshland, 2014*). These results suggest that desiccation may induce perturbations to membrane integrity and proteostasis. In nematodes, a partial loss of desiccation tolerance has also been observed with defects in protection against ROS (*Erkut et al., 2013*). Different hydrophilins have been reported to localize to different organelles, suggesting that unique organellar stress (i.e. mitochondria) may be an important feature of desiccation stress (*Candat et al., 2014*; *Hand et al., 2011*). These results led to the idea that desiccation causes a variety of lethal stresses, impacting multiple cellular constituencies, that require a host of specific stress effectors to mitigate.

We sought to identify stress effectors sufficient for desiccation tolerance, and to characterize their molecular functions. Here, using genetic assays with stationary and exponential phase cells, we demonstrate that trehalose and a single hydrophilin, Hsp12, unexpectedly synergize to completely alleviate all of the lethal damage that occurs due to severe water loss. We provide evidence that these two small molecules have both independent and synergistic activities to counter the misfolding and aggregation of protein reporters that occurs upon desiccation both in vivo and in vitro. We identify a novel in vitro role for Hsp12 as a membrane remodeler, an activity likely required to provide desiccation tolerance. Finally, our data suggest that despite sharing many of the same properties (high glycine content, small size, high hydrophilicity, disordered secondary structure), yeast hydrophilins clearly have distinct functions. These findings have profound implications for both stress biology and potential translational applications.

## Results

### Trehalose and Hsp12 are necessary and sufficient for desiccation tolerance

We reasoned that previous screens of the yeast deletion collection for sensitivity to short-term desiccation sensitivity were unproductive because likely more than one stress effector needed to be inactivated to significantly compromise tolerance. At least one of these factors was likely trehalose, given its role in long-term desiccation tolerance. Therefore, we started with a base strain that was unable to synthesize trehalose (*tps1Δ*) and introduced into it deletions for the remaining ~5000 non-essential genes in yeast (*Tong et al., 2001*). These double mutants were grown to stationary phase

and assayed for survival after short-term and long-term desiccation. We identified three groups with distinct phenotypes. The strains in the first group were inviable when dried for six days (*Supplementary file 2A2*, *Supplementary file 2B*). The strains in the second group were inviable even in the absence of stress (*Supplementary file 2A1*). The strains in the third group suppressed the long-term desiccation sensitivity of a *tps1Δ* (*Supplementary file 2A3*). While the strains in these last two groups identify potentially interesting genes for future study, we focused on the first group; the gene deletions in this group potentially inactivated candidate stress effectors besides trehalose that were needed for short-term desiccation tolerance.

One candidate gene from this group was *HSP12*. Hsp12 encodes one of 12 previously identified yeast hydrophilins and is the most highly expressed member of the family in stationary phase cells (*Garay-Arroyo et al., 2000*). To further investigate the potential synergism between Hsp12 and trehalose we compared quantitatively short- and long-term desiccation tolerance of stationary phase cells that were wild type, *tps1Δ*, *hsp12Δ* and *tps1Δhsp12Δ*. After a 30 day desiccation period, the viability of stationary phase wild type cells decreased only two-fold (*Figure 1A*). Although the viability of *tps1Δ* cells was similar to that of wild type cells after two days of desiccation, after the 30 day desiccation period, *tps1Δ* cell viability dropped more than 100-fold (*Figure 1A*). This result corroborated our previous evidence demonstrating a requirement of trehalose for long-term but not short-term desiccation tolerance (*Tapia and Koshland, 2014*). While *hsp12Δ* cells exhibited wild type levels of desiccation tolerance after both short- and long-term desiccation, the *tps1Δhsp12Δ* cells displayed a 100-fold drop in viability after only 2 days of drying. Furthermore, after 30 days of desiccation, no viable *tps1Δhsp12Δ* cells remained (*Figure 1A*). These results revealed that trehalose and Hsp12 act cooperatively to promote tolerance to both short- and long-term desiccation. Furthermore, although *tps1Δ* and *hsp12Δ* had both been shown to confer elevated sensitivity to heat stress, the *tps1Δhsp12Δ* double mutant was no more heat-sensitive than either single mutant alone (*Figure 1—figure supplement 1*) (*Gibney et al., 2015*; *Welker et al., 2010*). Thus, the protective synergism we observe between trehalose and Hsp12 is specific to desiccation stress.

We have shown previously that exponentially-growing cells with increased intracellular trehalose show a 1000-fold increase in desiccation tolerance relative to wild-type cells (*Tapia et al., 2015*). Hence, we tested whether exponentially dividing cells with engineered high-level expression of Hsp12 would also exhibit increased desiccation tolerance. Normally, Hsp12 is not highly expressed in unstressed dividing cells (*Praekelt and Meacock, 1990*). Therefore, we swapped its promoter with a strong constitutive promoter to drive Hsp12 expression in dividing cells to a level similar to that observed in stationary phase cells. These dividing cells exhibited a 1000-fold increase in desiccation tolerance, comparable to survival conferred by increased intracellular trehalose (*Figure 1B*). Thus, Hsp12, like trehalose, had a significant ability to mitigate one or more of the lethal stresses imposed by desiccation.

Next, we tested for potential synergy between trehalose and Hsp12 by examining the desiccation tolerance of exponentially-dividing yeast cells containing high levels of both factors. The desiccation tolerance of these cells was ~60 fold greater than cells that expressed only Hsp12 or trehalose. Furthermore, the absolute amount of tolerance (65–80% survival) was even greater than the tolerance of wild type cells in stationary phase (20–40% survival), the highest tolerance previously reported for yeast cells (*Figure 1B*) (*Calahan et al., 2011*). Our findings suggest that the functions of these two stress effectors must synergize to counter all the lethal effects of desiccation and subsequent rehydration on exponentially dividing yeast.

## Hsp12 can protect protein activity and aggregation of soluble cytosolic proteins during desiccation

Different in vitro experiments have suggested that trehalose helps prevent desiccation-induced proteotoxicity (*Kaushik and Bhat, 2003*; *Olsson et al., 2016*; *Singer and Lindquist, 1998*). With this in mind, we tested the ability of Hsp12 to modulate proteostasis under desiccation conditions. First, we examined the impact of Hsp12 on the in vitro activity of the well-established model proteostasis substrate citrate synthase (CS). We desiccated solutions of CS alone or with varying concentrations of Hsp12. These samples were rehydrated, and CS enzymatic activity was measured. Desiccation of CS caused an approximately 50% reduction in its enzymatic activity (*Figure 2A*). The presence of Hsp12 during drying maintained nearly all CS enzymatic activity in a concentration-dependent

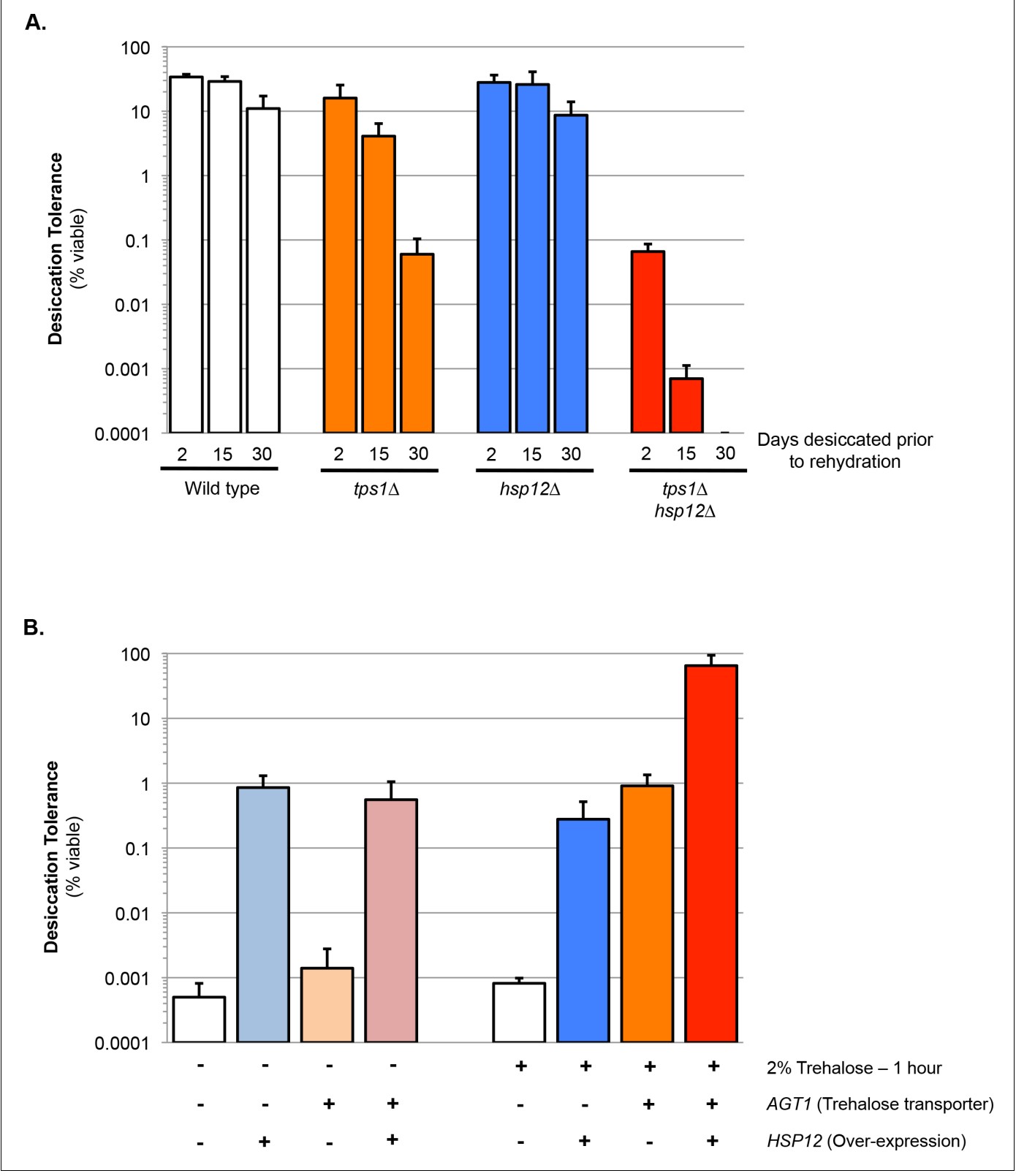

**Figure 1.** Trehalose and Hsp12 are necessary and sufficient for desiccation tolerance. (**A**) Yeast cells were grown to saturation (5 days), air-dried for 2, 15, or 30 days at 23°C, 60% relative humidity (RH), then rehydrated and assessed for viability by counting colony forming units (CFU). Desiccation tolerance of wild type, *tps1Δ*, *hsp12Δ* and *tps1Δhsp12Δ* cells. (**B**) Yeast cells (*nth1Δ*) were grown to mid-exponential phase (OD <0.5) in minimal media lacking histidine. Cells were then transferred to either minimal media lacking histidine (SC-His, 0% trehalose) or minimal media lacking histidine with

*Figure 1 continued on next page*

*Figure 1 continued*

trehalose (SC-His, 2% trehalose) for 1 hr. Cells were collected, washed, and air dried for 2 d at 23°C, 60% relative humidity (RH), then rehydrated and assessed for viability by counting colony forming units (CFU). Yeast cells are ± Trehalose transporter (*AGT1)* and ± Hps12 (p423-GPD-Hsp12).

DOI: https://doi.org/10.7554/eLife.38337.002

The following source data and figure supplement are available for figure 1:

**Source data 1.** Source data for *Figure 1A and B*.

DOI: https://doi.org/10.7554/eLife.38337.004

**Figure supplement 1.** Trehalose and Hsp12 heat tolerance.

DOI: https://doi.org/10.7554/eLife.38337.003

manner (*Figure 2A*). These results suggest that Hsp12 on its own can stabilize protein activity against desiccation in vitro.

We also examined the impact of Hsp12 on the in vivo activity of firefly luciferase (FFL) after desiccation. We expressed FFL in exponentially dividing wild type cells either expressing or not expressing Hsp12. Before desiccation, the levels of FFL luminescence were the same for all strains tested. After desiccation, luminescence in wild type cells not expressing Hsp12 was reduced by a factor of $10^5$ (*Figure 2—figure supplement 1A*). The presence of Hsp12 in these cells did not increase the activity of this very sensitive reporter (*Figure 2—figure supplement 1A*). Thus, Hsp12 clearly is not competent to stabilize the activity of all proteins against desiccation.

We next examined the ability of Hsp12 to protect against desiccation-induced aggregation in vitro and in vivo. CS was incubated and desiccated alone or with Hsp12. When dried alone, CS demonstrated high levels of aggregation, a 20-fold increase in its aggregation (*Figure 2B*). Increasing amounts of Hsp12 added before desiccation decreased aggregation of CS in a concentration-dependent manner (*Figure 2B*). Significant solubilization of CS occurred at a 2:1 ratio of Hsp12 to CS (*Figure 2B*, 225:120 µg/ml). This stoichiometry suggests this solubilization function of Hsp12 is not enzymatic, consistent with Hsp12s high expression during stress in vivo. Similarly, we desiccated exponentially dividing cultures of strains that expressed FFL either expressing or not expressing Hsp12. Upon rehydration, we prepared lysates of these cultures and subjected them to centrifugation to separate soluble FFL from insoluble FFL that had aggregated with itself or other insoluble proteins. For all strains tested, equivalent amounts of total FFL were present in cell lysates (*Figure 2—figure supplement 1B*). In the wild type strain not expressing Hsp12, no FFL was detected in the soluble fraction (*Figure 2—figure supplement 1B*). Increased Hsp12 expression significantly increased luciferase solubility. Thus, both in vivo and in vitro, Hsp12 helps mitigate desiccation-induced protein aggregation.

## Hsp12 proteostasis activities can act synergistically with, or independently of trehalose

Given the synergism between trehalose and Hsp12 to promote survival to desiccation, we used the CS and FFL assays to ask whether their activities in proteostasis also exhibited synergism. In vitro, trehalose, like Hsp12, increased the activity and solubility of CS after desiccation in a concentration dependent manner (*Figure 2E,F*). We looked for protective synergism of CS between trehalose and Hsp12 at suboptimal concentrations (*Figure 2—figure supplement 1C–D*). While trehalose and Hsp12 synergize to restore CS activity, limited synergistic protection against aggregation was observed. In vivo FFL activity was stabilized four-fold against desiccation in cells with high trehalose alone, but 14-fold in cells with both high Hsp12 and trehalose (*Figure 2C*). Thus, while Hsp12 had no effect on FFL activity on its own, it could synergize with trehalose to increase the activity of this reporter. Trehalose, like Hsp12, also increased the solubility of FFL after desiccation. Cells expressing both Hsp12 and trehalose further elevated FFL solubility (*Figure 2C,D*). These results demonstrate that Hsp12 and trehalose can modulate the activity and solubility of proteins independently or cooperatively depending upon the substrates, suggesting that Hsp12 and trehalose have both independent and dependent functions in proteostasis.

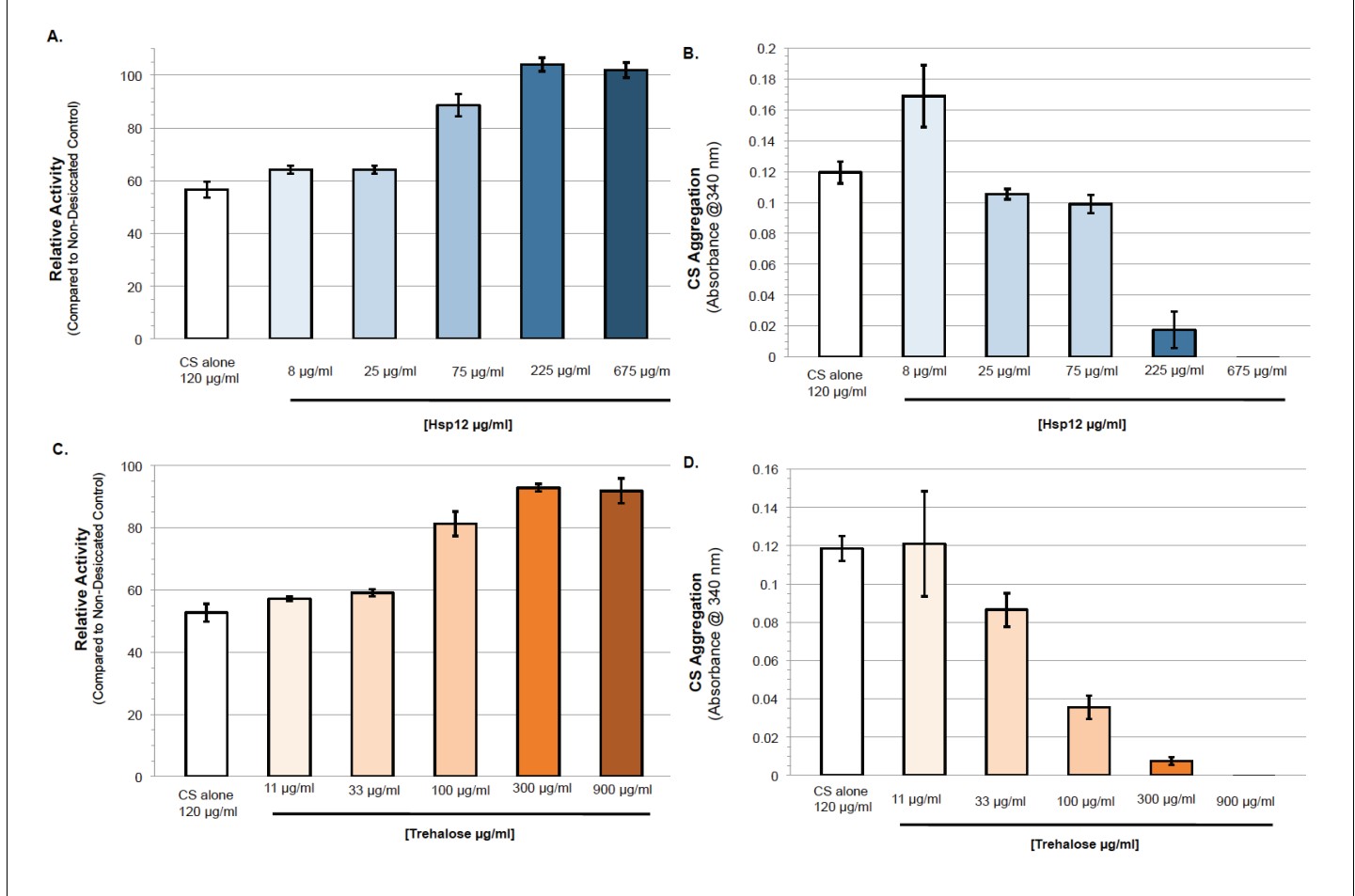

**Figure 2.** Hsp12 proteostasis activities can act synergistically with, or independent of trehalose. (A–B) (A) Enzymatic activity after drying of CS in presence of varying concentrations of Hsp12. CS enzymatic activity was measured by examining activity-induced changes in absorbance at 412 nm after drying compared to measurements before drying. (B) Citrate synthase desiccation-induced aggregation in the presence of varying concentrations of Hsp12. Absorbance at 340 nm. Citrate Synthase (0.12 mg). Absorbance measured after drying at 36°C for 24 hr followed by 5 hr at 23°C, then rehydration in water. Measurements after drying compared to measurements before drying. (C–D) (C) Enzymatic activity after drying of CS in presence of varying concentrations of trehalose. CS enzymatic was measured by examining activity-induced changes in absorbance at 412 nm after drying compared to measurements before drying. (D) Citrate synthase desiccation-induced aggregation in the presence of varying concentrations of trehalose. Absorbance at 340 nm. Citrate Synthase (0.12 mg). Absorbance measured after drying at 36°C for 24 hr followed by 5 hr at 23°C, then rehydration in water. Measurements after drying compared to measurements before drying.

DOI: https://doi.org/10.7554/eLife.38337.005

The following source data and figure supplement are available for figure 2:

**Source data 1.** Source data for *Figure 2A–D*.
DOI: https://doi.org/10.7554/eLife.38337.007

**Figure supplement 1.** Synergistic protection by trehalose and Hsp12
DOI: https://doi.org/10.7554/eLife.38337.006

## Hsp12 and trehalose act synergistically to propagate a membrane prion

Previously, we have used prion propagation as a sensitive measure of proteostasis (*Tapia and Koshland, 2014*). Prion propagation requires chaperones that promote oligomerization to generate new prions. Prion propagation also requires factors that prevent prion over-oligomerization. If the prion aggregate becomes too large, it fails to transfer into daughter cells during cell division, resulting in curing over time. Previously, we showed that [*PSI⁺*], a cytoplasmic prion, or [*GAR⁺*], a membrane-associated prion, are propagated efficiently between desiccated stationary phase cells and their

daughter cells that form upon rehydration in growth media (*Tapia and Koshland, 2014*). This propagation is greatly impaired if the desiccated cells lack trehalose.

Unlike *tps1Δ* cells, *hsp12Δ* cells were able to propagate [*PSI*⁺] with or without desiccation, suggesting that trehalose, but not Hsp12, is required to prevent [*PSI*⁺] hyper-aggregation (*Figure 3A*). This distinction provides another example of independent function of Hsp12 and trehalose in the proteostasis of cytosolic proteins. However, our analysis of the propagation of the membrane prion [*GAR*⁺] revealed a striking phenotype in the *tps1Δhsp12Δ* double mutant. Even in the absence of desiccation, the percentage of [*GAR*⁺] cells in *tps1Δhsp12Δ* cultures was reduced almost 500-fold relative to wild type or *hsp12Δ* cells, and 50-fold relative to *tps1Δ* (*Figure 3B*). Thus, Hsp12 and trehalose promote propagation of this membrane prion even under aqueous conditions. After desiccation, the percentage of [*GAR*⁺] cells was unchanged in *hsp12Δ* mutant cells but was undetectable in both the *tps1Δ* and *tps1Δhsp12Δ* cells (*Figure 3B*). The dramatic impact of *tps1Δ* alone on [*GAR*⁺] propagation after desiccation likely masked the additional impact of the loss of Hsp12.

## Hsp12 remodels lipid vesicles

The synergistic function of trehalose and Hsp12 in [*GAR*⁺] propagation could be due to their direct effect on membrane proteins and/or an indirect effect on overall membrane structure and integrity. Intriguingly, prior studies indicate that trehalose stabilizes membranes in vitro and this stabilization may prevent disruptive phase transitions that occur upon rehydration (*Crowe et al., 1984*, *1987*; *Erkut et al., 2011*; *Leslie et al., 1994*). Hsp12 also has been reported to increase membrane stability (*Sales et al., 2000*; *Welker et al., 2010*). An alternative/additional membrane function for Hsp12 was suggested by the fact the addition of certain lipids induce the transition of Hsp12 from a disordered state to one with secondary structure (*Herbert et al., 2012*; *Singarapu et al., 2011*; *Welker et al., 2010*). Secondary structure prediction and NMR data identify four alpha-helical regions in Hsp12, with each of the three larger helices having the features of an amphipathic helix (*Herbert et al., 2012*; *Singarapu et al., 2011*; *Welker et al., 2010*). Numerous membrane-remodeling proteins use amphipathic helices, embedding the hydrophobic face partway into the bilayer to shape membranes (*Boucrot et al., 2012*; *Meinecke et al., 2013*). Given these considerations, we hypothesized that Hsp12 might have membrane remodeling activities.

One readout of membrane remodeling activity is the capacity to vesiculate large liposomes into small vesicles (*Boucrot et al., 2012*). For example, proteins known to be involved in endocytic membrane trafficking lead to the vesiculation of liposomes (*Boucrot et al., 2012*). Therefore, we tested whether addition of Hsp12 to dimyristoylphosphatidylglycerol (DMPG) liposomes would generate nanovesicles. DMPG liposomes incubated in the absence of Hsp12 sedimented into the pellet fraction due to their large size (*Figure 4A*). By contrast, when incubated in the presence of Hsp12, DMPG appeared in the supernatant fraction (*Figure 4A*). Liposome vesiculation mediated by Hsp12 was incredibly robust and occurred even at low concentrations (Hsp12 - 1.5 µg/ml), with complete vesiculation occurring in a concentration-dependent manner (*Figure 4B*). To assess vesiculation directly, we examined these same samples by electron microscopy. Prior to the addition of Hsp12, liposomes exhibited expected spherical shapes and sizes (<200 nm). After the addition of Hsp12, very few intact liposomes were observed, replaced by membrane remnants (*Figure 4C*). Trehalose alone did not have any vesiculation activity nor did it inhibit the effect of Hsp12 on liposomes under the conditions tested (*Figure 4A*). These results show that Hsp12, but not trehalose, exhibits membrane remodeling activity in vitro and suggest that their synergistic impact on [*GAR*⁺] propagation may result from different biophysical activities.

## Desiccation protection is not conserved among all hydrophilins

In previously reported in vitro assays, different hydrophilin functions appear to be interchangeable (*Chakrabortee et al., 2007*; *Goyal et al., 2005*; *Hand et al., 2011*; *Liu et al., 2011*). Their interchangeability has led to models for a common function (for example, protein coating) based upon simple biophysical properties dictated by their shared primary sequence features of charge and glycine richness, and their unstructured state. This common function would explain the previously reported minor changes in desiccation sensitivity upon inactivation of individual hydrophilins, as their redundant functional properties should be present in all hydrophilins. To address possible functional redundancy, a single strain lacking all non-essential yeast hydrophilins was engineered (*8XΔ: hsp12Δ*

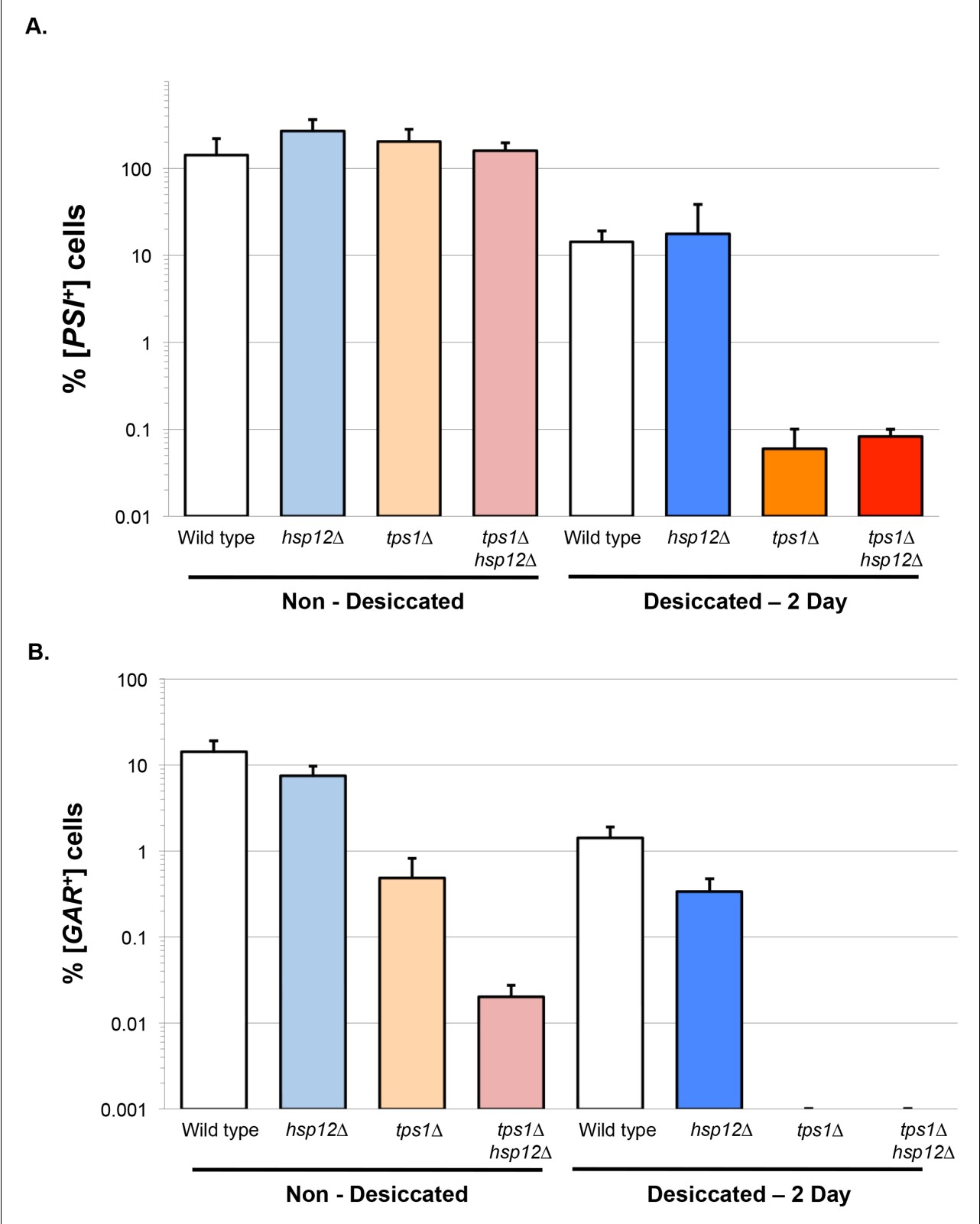

**Figure 3.** Hsp12 and trehalose can act synergistically in membrane proteostasis. (A–B) Prion propagation as a measure of in vivo protein propagation after desiccation. (A) [PSI+] prion propagation before, and after desiccation. To assess [PSI+] prion state, cells were plated on media lacking adenine (SC-ADE) compared to non-dried controls. (B) [GAR+] prion propagation before, and after desiccation. To assess [GAR+] prion state, cells were plated on YP media with 2% glycerol and 0.05% glucosamine compared to non-dried controls.

*Figure 3 continued on next page*

*Figure 3 continued*

DOI: https://doi.org/10.7554/eLife.38337.008

The following source data is available for figure 3:

**Source data 1.** Source data for *Figure 3A and B*.

DOI: https://doi.org/10.7554/eLife.38337.009

*gre1Δ sip18Δ stf2Δ nop6Δ ybr016wΔ yjl144wΔ ynl190wΔ*) (*Suzuki et al., 2011*). However, even when lacking all non-essential hydrophilins, stationary phase yeast cells still demonstrated robust desiccation tolerance (*Figure 5A*). This result suggested that the lack of a phenotype was not due to hydrophilin redundancy. Indeed, we could not have observed a strong synergism between *tps1Δ* and *hsp12Δ* mutations if other hydrophilins could substitute for the missing Hsp12.

To address more directly whether the remarkable features of Hsp12 were common to other yeast hydrophilins, we studied a second yeast hydrophilin, Stf2, that is also expressed in stationary phase cells (*Garay-Arroyo et al., 2000*). Expressing high levels of Stf2 in exponentially-dividing cells did not significantly improve desiccation tolerance and failed to provide any synergistic tolerance with trehalose (*Figure 5B*, *Figure 5—figure supplement 1A,B*). Additionally, deletion of Stf2 alone (*stf2Δ*), or in combination with a loss of trehalose synthesis (*tps1Δstf2Δ*) had no effect on the short-term desiccation tolerance of stationary phase yeast, unlike the pronounced increase in sensitivity displayed by *tps1Δhsp12Δ* cells (*Figures 1A* and *5B*). Unlike Hsp12, Stf2 also did not exhibit any detectable secondary structure in the presence or absence of DMPG (*Figure 5C*, Hsp12 + DMPG demonstrates classical alpha-helical CD signature). Stf2 also fails in vesiculating membranes (*Figure 5D–E*). These functional differences between Hsp12 and Stf2 suggest that hydrophilins likely carry out distinct biological and molecular functions, despite sharing general physical properties. Moreover, the biological and biochemical differences between Hsp12 and Stf2 further support the view that the specific membrane remodeling activity of Hsp12 contributes to its desiccation tolerance-promoting function.

## Discussion

We exploited the conditional desiccation tolerance of yeast to provide important new insights into the stress effectors of desiccation tolerance. While previous studies in yeast, nematodes and tardigrades have examined the impact of individual hydrophilins or trehalose on desiccation tolerance, no in vivo studies have looked at the combined effect of loss or gain of both trehalose and a hydrophilin. Here, we demonstrate that stationary phase yeast cells lacking both trehalose and Hsp12 (*tps1Δhsp12Δ*) exhibit a greater than 100-fold loss in short-term desiccation tolerance and 10,000-fold loss in long-term desiccation. The magnitude of the desiccation sensitivity suggests that these two factors are critical to mitigate most of the lethal stresses associated with desiccation. In addition, we show that supplementing cells with only trehalose and Hsp12 was sufficient to make desiccation-sensitive dividing cells desiccation tolerant, this tolerance is higher than that which is observed naturally in stationary phase cells. This increase in desiccation tolerance was far greater than cells expressing each factor alone, again suggesting a synergistic interaction to suppress desiccation-induced stresses. In summary, while dividing cells have mechanisms to deal with the stresses imposed by subtler changes in available water; the new or increased amplitude of stresses imposed by severe/complete water loss can all be mitigated by trehalose and Hsp12 alone.

Hsp12 had not previously been implicated in desiccation tolerance. Rather, Hsp12 had been reported to mitigate lethality due to heat stress and osmolarity (*Welker et al., 2010*). We were unable to reproduce those findings. Furthermore, the heat stress that was used in the previous study was 58°C, a temperature that budding yeast is unlikely to experience in the wild or in fermenting vats. In contrast, desiccation occurs commonly in nature, so the desiccation tolerance conferred by Hsp12 is likely one of its physiological functions. Our results, coupled with the previous demonstration for the importance of individual hydrophilins for desiccation tolerance in tardigrade and nematodes provide compelling evidence for the causal roles of a hydrophilin in desiccation tolerance in all anhydrobiotes (*Boothby et al., 2017*; *Erkut et al., 2013*).

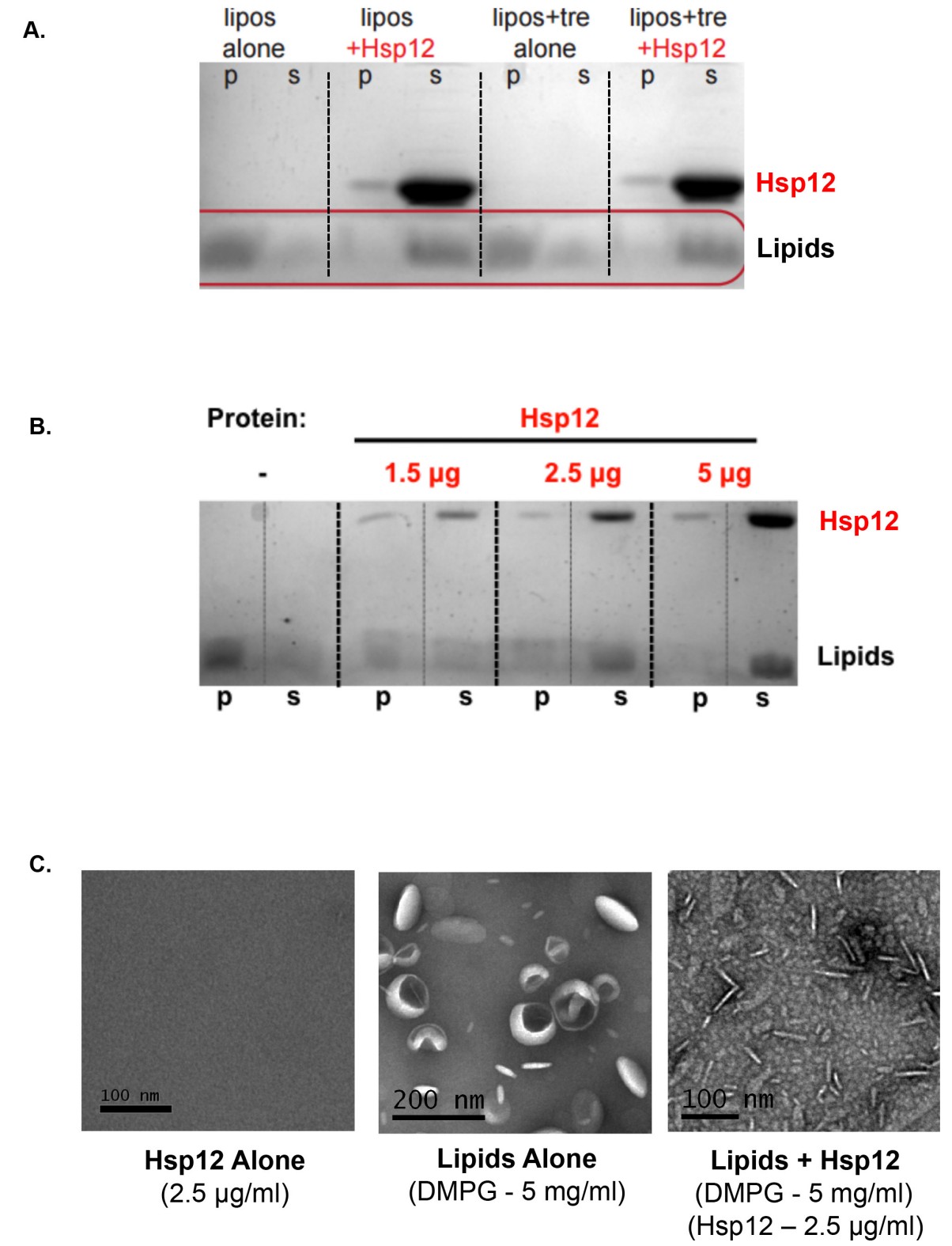

**Figure 4.** Hsp12 causes membrane remodeling. (**A**) DMPG liposomes (5 mg/ml) where incubated for 1 hr at room temperature in the presence or absence of Hsp12 (25 μg/ml) and/or in the presence or absence of trehalose (2% final). Pellet (p) and supernatant (s) fractions where separated by high-speed centrifugation, and lipid distribution was assessed by SDS-PAGE followed by altered Coomassie staining (10% acetic acid only). (**B**) DMPG liposomes (5 mg/ml) where incubated for 1 hr at room temperature in the presence or absence of Hsp12 (1.5–5 μg/ml). Pellet (p) and supernatant (s)

*Figure 4 continued on next page*

*Figure 4 continued*

fractions where separated by high-speed centrifugation, lipid distribution was assessed by SDS-PAGE followed by altered Coomassie staining (10% acetic acid only). (C) Samples for electron microscopy were prepared the same as in A, with Hsp12 at 2.5 µg/ml. Images where taken from samples prior to centrifugation. Samples where spread on glow-discharged EM grids and stained using 2% uranyl acetate.

DOI: https://doi.org/10.7554/eLife.38337.010

The following source data is available for figure 4:

**Source data 1.** Source data for *Figure 4B*.
DOI: https://doi.org/10.7554/eLife.38337.011
**Source data 2.** Source data for *Figure 4C*.
DOI: https://doi.org/10.7554/eLife.38337.012

Surprisingly, the important role of Hsp12 in desiccation tolerance appears to reflect a unique function amongst the hydrophilin family based on several considerations. First, if any of the other 11 hydrophilins in yeast had the same activity as Hsp12, then an *hsp12Δ* strain would not show synergistic desiccation sensitivity with *tps1Δ*, and *hsp12Δ* would not have been picked up in our screen for trehalose desiccation synthetic-sensitivity. Second, desiccation sensitivity was not observed in a strain deleted for Hsp12 and all the other seven non-essential hydrophilin genes, again suggesting the absence of functional redundancy between hydrophilins. Third, further analysis of one of these eight hydrophilins, Stf2, showed that it displayed none of the synergism for desiccation tolerance or sensitivity with the presence or absence of trehalose that we observe with Hsp12. A previous report suggested that *stf2Δ* demonstrated a subtle 5-fold loss in desiccation tolerance that we do not observe; this previous study rehydrated cells with warmed media introducing a heat shock variable wherein the function of Stf2 might be required (*López-Martínez et al., 2012*). The apparent uniqueness of Hsp12 as a critical factor for desiccation tolerance challenges the notion that all hydrophilins belong to a redundant family of proteins with common biological functions. Indeed, the uniqueness of Hsp12 may be the tip of the iceberg in which studies of the other hydrophilins in yeast will reveal unique functions in different aspects of cell biology.

What is the molecular function of Hsp12 underlying its distinct role in desiccation tolerance? Several studies have been reported the ability of hydrophilins in vitro to protect against protein aggregation and protein misfolding (*Goyal et al., 2005*; *Hincha and Thalhammer, 2012*). However, evidence for the in vivo relevance of these in vitro findings was lacking. Here, we show that Hsp12 demonstrates both of these activities with analyzed substrates both in vitro and in vivo. Specific protein substrates of Hsp12 may be critical for organismal desiccation tolerance. Support for this idea will require identifying such key substrates and testing their protection by Hsp12. However, we show that the in vivo ability of Hsp12 to promote desiccation tolerance correlates with the ability of Hsp12 to acquire secondary structure in the presence of lipids and to cause membrane vesiculation. This membrane remodeling activity is consistent with the requirement of Hsp12 and trehalose to allow propagation of the membrane prion [*GAR*+]. Given the impact of Hsp12 and trehalose on [*GAR*+] prion propagation in aqueous conditions, Hsp12's vesiculation function may play an important role during rehydration.

The synergistic biological interactions of trehalose and Hsp12 in desiccation tolerance may reflect their ability to modulate common stresses by different yet complementary mechanisms. As suggested by in vitro experiments, the protective effect of trehalose on cytosolic proteins may result from its ability to form an amorphous liquid glass inside cells in the absence of water, while Hsp12's protective effect may result from aiding in the formation and strength of trehalose in its vitrified state. Additionally, trehalose has also been proposed to protect against severe membrane damage during desiccation. Damage that escapes protection by trehalose may be removed by Hsp12's vesiculation function. Interestingly, it has recently been noted that intrinsically disordered proteins and protein regions have been shown to contribute to phase transitions leading to a more ordered cell (*Alberti and Hyman, 2016*; *Mitrea and Kriwacki, 2016*; *Uversky et al., 2015*). Hsp12 might protect proteins from aggregating via means of phase separating susceptible proteins and hence buffering the cytosol. Additional genetic and biochemical experiments will be needed to test these models.

Hsp12's lipid-induced folding, its in vitro membrane remodeling activity and its in vivo role in desiccation tolerance are not shared with Stf2, another yeast hydrophilin. These apparently unique

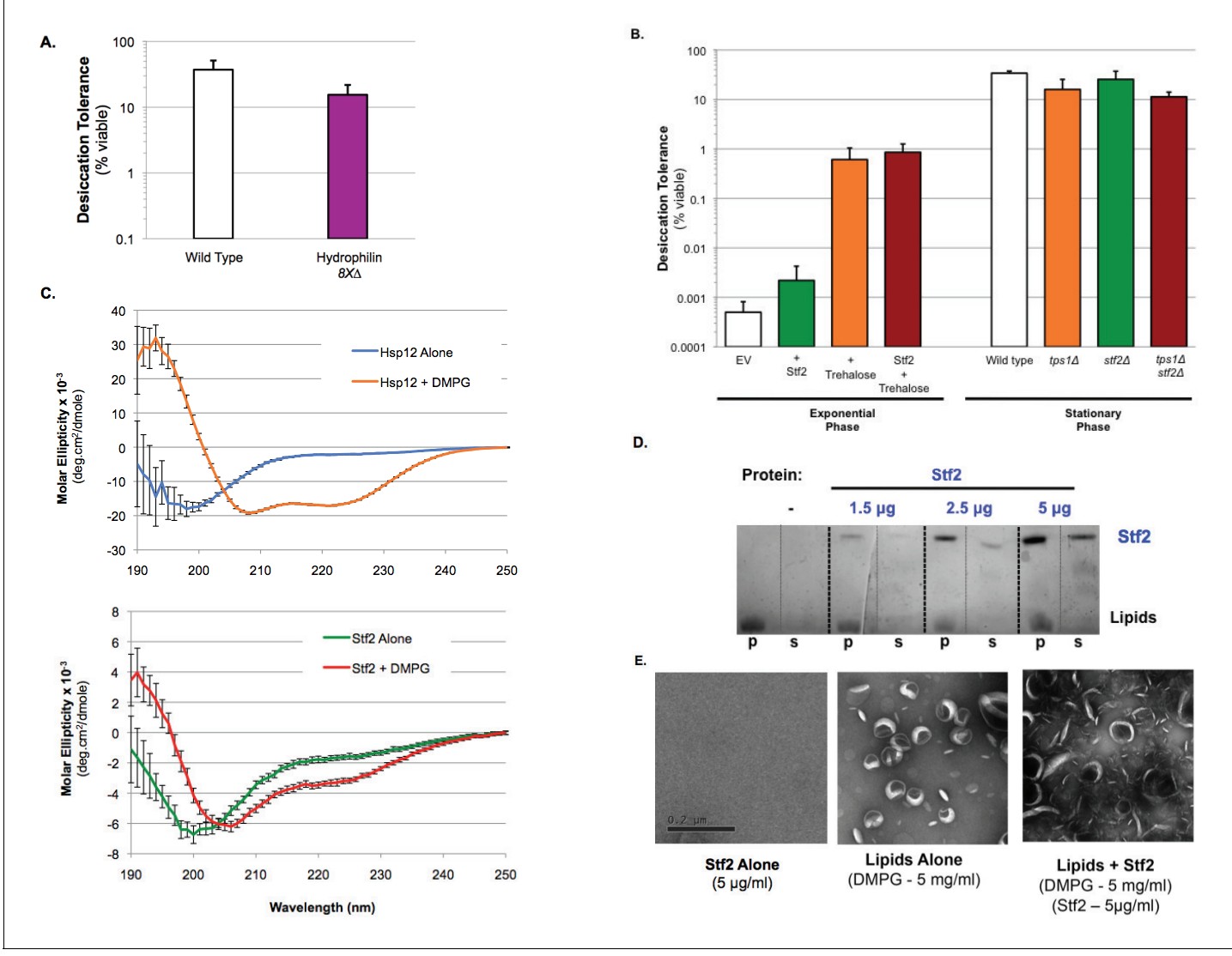

**Figure 5.** Desiccation protection not a common hydrophilin feature. (**A**) Yeast cells were grown to saturation (5 days), air-dried for 2 days at 23°C, 60% relative humidity (RH), then rehydrated and assessed for viability by counting colony forming units (CFU). Desiccation tolerance of wild type vs. *8XΔ*: *hsp12Δ gre1Δ sip18Δ stf2Δ nop6Δ ybr016wΔ yjl144wΔ ynl190wΔ* cells. (**B**) Exponential Phase. Yeast cells (*nth1Δ, AGT1+*) were grown to mid-exponential phase (OD <0.5) in rich media (YPD). Cells were then transferred to either rich media (YPD, 0% trehalose) or rich media with trehalose (YPD, 2% trehalose) for 1 hr. Cells were collected, washed, and air dried for 2 d at 23°C, 60% relative humidity (RH), then rehydrated and assessed for viability by counting colony forming units (CFU). Stationary Phase. Yeast cells are ± AGT1 (trehalose transporter) and ± Stf2 (over-expression from 2μ plasmid, GPD promoter). Yeast cells were grown to saturation (5 days), air-dried for 2 days at 23°C, 60% relative humidity (RH), then rehydrated and assessed for viability by counting colony forming units (CFU). Desiccation tolerance of wild type, *tps1Δ*, *stf2Δ* and *tps1Δstf2Δ* cells. (**C**) Circular dichroism spectroscopy performed on 0.32 mg/ml Hsp12 or 0.32 mg/ml of Stf2 in the presence or absence of 1.2 mg/ml DMPG small unilamellar vesicles (SUVs), measuring from 250 to 190 nm. Units converted to Mean Molar Residue Ellipticity, accounting for concentration and protein size. (**D**) DMPG liposomes (5 mg/ml) where incubated for 1 hr at room temperature in the presence or absence of Stf2 (1.5–5 μg/ml). Pellet (p) and supernatant (s) fractions where separated by high-speed centrifugation, lipid distribution was assessed by SDS-PAGE followed by altered Coomassie staining (10% acetic acid only). (**D**) Samples for electron microscopy where the same as in A, with Stf2 at 5 μg/ml. Images where taken from samples prior to centrifugation. Samples where spread on glow-discharged EM grids and stained using 2% uranyl acetate.

DOI: https://doi.org/10.7554/eLife.38337.013

The following source data and figure supplement are available for figure 5:

**Source data 1.** Source data for *Figure 5A–C*.
DOI: https://doi.org/10.7554/eLife.38337.015
**Source data 2.** Source data for *Figure 5D*.
DOI: https://doi.org/10.7554/eLife.38337.016

*Figure 5 continued on next page*

eLIFE Research article

Biochemistry and Chemical Biology | Cell Biology

*Figure 5 continued*

**Source data 3.** Source data for *Figure 5E*.
DOI: https://doi.org/10.7554/eLife.38337.017
**Figure supplement 1.** Hsp12 and Stf2 are expressed to similar levels in growing cells.
DOI: https://doi.org/10.7554/eLife.38337.014

features of Hsp12 provide new insights into hydrophilins and stress biology. Like yeast, most organisms express a large family of hydrophilins genes. However, the reason why hydrophilin families are so large remains a mystery. The distinct causal role of Hsp12 in desiccation tolerance reveals that the generic properties of hydrophilins of charge, glycine composition and disorder are not sufficient to mitigate the stresses of desiccation. Thus, other hydrophilins, like Stf2, must have other distinct functions. One simple idea is that these other functions mitigate yet to be determined cellular processes or different stresses.

We provide in vivo and in vitro evidence that both trehalose and Hsp12 can modulate proteostasis in response to desiccation. We show that in vivo they prevent desiccation-induced aggregation of firefly luciferase and of citrate synthase in vitro. Trehalose and Hsp12 also synergize, albeit weakly, to protect firefly luciferase activity in vivo, and individually they protect citrate synthase activity in vitro. Whether the stabilization of enzyme activity is different or just another manifestation of their anti-aggregation activity is not clear. Interestingly, the proteostasis functions of trehalose and Hsp12 are not identical as cells lacking only trehalose are compromised for [*PSI*$^+$] and [*GAR*$^+$] propagation while lacking cells Hsp12 alone are not. These differences likely indicate that trehalose and Hsp12 may modulate proteostasis by different mechanisms.

We propose a working model based upon the requirements of Hsp12 and trehalose for desiccation tolerance and observations reported previously. Trehalose prevents abnormal membrane vesicles in desiccated nematodes (*Erkut et al., 2011*). In vitro experiments suggest that trehalose intercalates into membranes to alter their melting properties (*Crowe et al., 1984*, *1987*; *Leslie et al., 1994*). Based upon these observations we propose that desiccation induces membrane damage that can be prevented by intercalation of trehalose. If damage occurs, it could then be removed by Hsp12's remodeling activity. These distinct activities provide a possible explanation for the synergistic requirement for trehalose and Hsp12 in both short- and long-term desiccation tolerance.

The finding that trehalose and Hsp12 together are sufficient to mitigate the major stresses of desiccation has major implications for engineering desiccation/drought tolerance in other organisms. The ability to generate desiccation tolerance with only two factors makes this engineering eminently more feasible and approachable from an evolutionary perspective. Attempts to engineer plants to synthesize more trehalose have met with technical difficulties because of the disaccharide's additional roles in metabolism. Our results provide new impetus to overcome those hurdles. Furthermore, manipulation of Hsp12 expression has no known metabolic side effects, at least in budding yeast. Given the sufficiency of Hsp12 alone to confer partial desiccation tolerance, it will be very interesting to test whether Hsp12 by itself might confer drought tolerance to plants or other organisms that might benefit from this remarkable trait.

## Materials and methods

### Strains and growth conditions

Standard yeast propagation and transformation procedures were used. Yeast strains are described in *Supplementary file 3*. Strains were grown in nonselective (YP, 1% yeast extract and 2% peptone) or selective (synthetic complete, SC) media containing 2% glucose lacking specific selectable amino acids. Cultures were grown to saturation from a single colony by incubating cultures 5 days at 30°C for stationary phase experiments. Cultures where grown to an OD – 0.5 for mid-log experiments. All experiments were repeated at least two times on separate days with separate isolates when appropriate.

## Desiccation tolerance assay

Saturated Cultures: Approximately $10^7$ cells were withdrawn from liquid cultures and washed twice in dilute water and then brought to a final volume of 1 ml. Undesiccated controls were plated for colony counting. Two hundred microliter aliquots were then transferred to a 96-well tissue culture plate (Becton Dickinson, 353075) and centrifugated, and water was removed without disturbing the cell pellet. Cells were allowed to desiccate in a 23°C incubator with a constant 60% relative humidity (RH), with the lid raised, for at least 48 hr. Long-term desiccation experiments were kept for indicated time periods in a 96-well tissue culture plates at 23°C, 60% RH. Samples were resuspended in assay buffer and plated for colony counting.

Logarithmic Samples: Cells were grown to midexponential phase (OD <0.5) in selective media, depending on strain and plasmid necessities. Cells were then transferred to inducible media: ±2% trehalose, for 1 hr. Following induction, ~$10^7$ cells were withdrawn from liquid cultures and same parameters for drying where used as with saturated cultures. Data were entered into a spreadsheet (Microsoft Excel 2008 for Mac version 12.3), and the number of colony forming units per milliliter (cfu/mL) for each plate was computed. For each experiment, number of colony forming units per milliliter for the two controls was averaged. The relative viability of each of the two experimental samples was determined by dividing the number of colony forming units per milliliter for that sample by the average number of colony forming units per milliliter of the control plates. These two relative viability values were then averaged and their SD was computed using the STDEVP worksheet function.

## Luciferase assay

Yeast cells (nth1Δ, AGT1 strains) (±2% Trehalose, ±Hsp12), bearing 2μ plasmids that direct the expression of a temperature-sensitive firefly luciferase-fusion protein from the constitutive glyceraldehyde-3-phosphate (GPD) promoter (p426-GPD-FFL) or empty vector (pEV, p426-GPD), were grown to mid-log phase in SC –His - Ura. Luciferase activity was measured in vivo by addition of 0.5 mM D-Luciferin (Sigma) to equal number of intact cells. Light emission was measured immediately with a TD-20/20 Luminometer (Turner Designs, Sunnyvale CA). Desiccated samples were air dried for two days followed by rehydration in SC-His - Ura + cycloheximide (10 μl/ml, to block new FFL protein synthesis) and subjected to the same assay treatment after rehydration. Measurements are reported as Relative Light Units. Using glass bead lysis, total cellular protein was extracted. After removing unbroken cells by low-speed centrifugation (1000 g for 3 min), cleared lysates were spun at high speed (350,000 g for 10 min, Beckman Coulter TLA-100) to collect insoluble aggregates. Total protein from cleared lysates and high-speed supernatants were then followed by SDS-PAGE and reacted with antiserum recognizing luciferase. All experiments were repeated three times on separate days with separate isolates.

## Citrate synthase assay

### Citrate synthase preparation

Citrate Synthase from Porcine Heart ammonium sulfate suspension (C3260) was purchased from Sigma Aldrich. Suspension was centrifugated at 14,000 rpm for 10 min at 4°C. Supernatant was discarded and the pellet was resuspended in 750 μl MilliQ water. To remove residual ammonium sulfate, citrate synthase was desalted in MilliQ water using a 5 ml HiTrap Desalting Column (GE Healthcare). Fractions were evaluated for protein content via staining 30 μl of each fraction with 100 μl of Coomassie protein assay reagent (Sigma) diluted 1:5 in MilliQ water. Fractions with detectable protein content were pooled and concentrated using an Amicon Ultra 0.5 ml 10K centrifugal filter unit (Merck Millipore Ltd). Final concentration determined via measuring absorbance at 280 nm on a Nanodrop 2000 spectrophotometer (Thermo Scientific).

### Citrate synthase aggregation and enzymatic activity assay

Citrate synthase (CS) and protectants were added to MilliQ water to achieve final concentration of 0.12 mg/ml CS, and varying concentrations of protectants in 100 μl.

For measuring enzymatic activity, 1 μl of each sample was diluted by a factor of 17 in MilliQ water. 2 μl of this dilution was then mixed with 198 μl of Enzymatic Activity Assay Solution [100 mM oxaloacetic acid (Sigma-Aldrich), 100 mM 5,5′-Dithiobis(2-nitrobenzoic acid) (Sigma-Aldrich), 150 mM Acetyl Coenzyme A (Sigma-Aldrich)] and absorbance at 412 nm was measured immediately and

after 120 s in a 50 µl Micro Cell cuvette (Beckman Coulter) with measurements made with a 720 UV/ Vis spectrophotomer (Beckman Coulter) equipped with a 50 µl Micro Cell adapter (Beckman Coulter). After measurement, samples were moved to 1.5 ml microcentrifuge tubes and set to dry in a Centrivap Concentrator (Labconco) set to 35°C and 29 bar for 24 hr. Samples were then moved to a 23°C incubator and allowed to incubate open to atmospheric pressure for an additional 5 hr. Samples were rehydrated in 100 µl MilliQ water, resuspended vigorously 5 min after addition of water. 2 µl of this resuspension was added to 198 µl Enzymatic Activity Assay Solution, mixed and moved to a cuvette, measuring absorbance at 412 nm immediately and after two minutes. Change in absorbance across two minutes after drying compared to before drying to achieve relative activity. For measuring aggregation, absorbance at 340 nm was measured for each sample before and after drying. Drying and rehydration performed as described for enzymatic activity assay. Measurements after drying subtracted from before drying measurements to achieve change in aggregation after drying. Cuvette was washed with MilliQ water and 95% ethanol between all measurements.

## Expression and purification of Hsp12p

BL21 *E. coli* cells transformed with a pET-15B vector with HSP12 behind a galactose-inducible promoter grown overnight in 30 ml cultures in LB with 100 ng/ml ampicillin. Transformants were grown up overnight in 30 mL cultures in LB with 100 ng/ml ampicillin at 37°C. 20 mL of this culture were added to 2L of LB with ampicillin and incubated at 37° C until culture reached an OD600 of 0.5–0.9, as determined by spectrophotometry using a 720 UV/Vis spectrophotomer (Beckman Coulter). HSP12 expression induced by addition of IPTG (Isopropyl β-D-1-thiogalactopyranoside) to final concentration of 1 mM and incubation for an additional 4 hr at 37°C. Cultures were then pelleted by centrifugation at 4000 rpm for 30 min at 4°C. Discarded supernatant and resuspended cells in 50 mL of 1x PBS. Cells were frozen with liquid nitrogen, and stored at −80°C until use. Pellets were isolated and resuspended in lysis buffer (2 mM EDTA, 2 mM DTT, 20 mM HEPES, pH 7.4) then sonicated using a blunt tip sonicator set to 30% amplitude, 20 min, pulse 1 s on, 1 s off at 4°C. Sonication-lysed cells were then centrifugated and supernatant was collected. This sample was then boiled at 95°C for 15 mins and immediately moved to ice. A pale yellow precipitate was separated away from the proteins that are still soluble after boiling by centrifugation. Supernatant was then applied to a HiTrap Capto-Q ImpRes anion exchange column (GE Healthcare Life Sciences), and washed with lysis buffer with increasing concentrations of NaCl. Hsp12 was eventually eluted by buffer between 7.5 and 10 mM NaCl. The protein was then concentrated using a MW3000 filtration unit, aliquoted, frozen in liquid nitrogen, and stored at −80°C until use (*Supplementary file 1*).

## Expression and purification of Stf2p

STF2 fused to GST was cloned into a pGEX6 vector behind a galactose-inducible promoter and transformed into BL21 cells. Transformants were grown overnight at 37°C in 30 mL of LB liquid media, selecting for plasmid with 75 ng/ml ampicillin. 2 L LB cultures with 75 ng/ml ampicillin were inoculated with 20 ml of these overnight cultures and set to 37°C and 140 rpm in an Innova 4330 incubator (New Brunswick Scientific). After 4 hr, expression of STF2 was induced with 4 mL of 0.1M IPTG and cultures were allowed to incubate overnight at 18°C. Cells were harvested and lysed in lysis buffer 2 (150 mM NaCl, 2 mM EDTA, 2 mM DTT, 20 mM HEPES, pH 7.4) using the same sonication method as above (see Purification of Hsp12). Then, lysed cells were centrifugated and the supernatant was collected. 1.5 ml of Pierce Glutathione Agarose beads from Thermo Fisher Scientific were added and incubated in the supernatant for an hour at 4°C. Beads were washed in 3 washes in lysis buffer 2, 3 washes in lysis buffer 3 (500 mM NaCl, 2 mM EDTA, 2 mM DTT, 20 mM HEPES, pH 7.4), and 3 washes in lysis buffer 2, and resuspended in lysis buffer 2. Prescission protease was added to washed beads and this mixture was incubated at 4°C overnight. The mixture was then centrifugated and the supernatant was collected, concentrated with an Amicon Ultra-4 Ultracel 3K centrifugal filter unit from Millipore, alliquoted, frozen in liquid nitrogen, and stored at −80°C (Supplementary Figure 1).

## Prion phenotypic assays

To assess the prion state, cells were plated on media lacking adenine (SC-ADE) for [*PSI*+] strains, or on YP media with 0.05% Glucosamine and 2% Glycerol for [*GAR*+] growth was compared to their

ability to grow on rich YPD/GAL media. To determine % Prion = 100 X CFU Desiccated Cells (Prion Selective Media)/CFU Desiccated Cells (Non-Selective Rich Media).

## Liposome preparation and vesiculation

DMPG liposomes (Avanti Polar Lipids (840445) in 100 mM NaCl, 20 mM Tris-Hcl pH 7.4 were sonicated for 2 min in a water bath at room temperature. Liposomes were incubated with protein for 60 min and centrifugated at 250,000 g for 15 min in a Beckman Coulter TLA-100 rotor. Resuspended pellets and supernatants were analyzed by SDS-PAGE. Gels were stained with 0.1% Coomassie in 10% Acetic acid and destained in water. For EM, samples were spread prior to centrifugation on glow-discharged electron microscopy grids and stained using 2% uranyl acetate and viewed in a Joel 1200 EX Transmission Electron Microscope.

## Preparing liposomes for circular dichroism

1,2-dimyristoyl-*sn*-glycero-3-phospho-(1'-rac-glycerol) (DMPG) from Avanti Polar was stored in chloroform at −80°C in glass tubes until used. DMPG aliquots were thawed and aspirated using a stream of nitrogen gas to form a clear, dry film on the inside surface of the tube. Tubes were then placed inside a vacuum desiccator and sealed. Vacuum was then applied to the desiccator, and the samples were left at room temperature under vacuum overnight to remove any remaining chloroform. Then, they were resuspended in 10 mM sodium phosphate, pH 7.4 and moved to a 1.5 ml plastic microcentrifuge tube. Resuspended aliquots were then vortexed until a homogenous, cloudy white solution was formed. Tubes were placed on ice in a 4°C room, and sonicated using a probe tip sonicator equipped with a blunt tip and set to 20% amplification, 7.5 min, with pulses of 2 s on and 2 s off. Lipids were then spun down at 14,000 rpm at 4°C for 10 min to precipitate any metal shards from sonication. Supernatant was then moved to a new 1.5 ml microcentrifuge tube and left on ice until use in circular dichroism.

## Circular dichroism spectra

Circular Dichroism (CD) spectra were obtained as previously described with some adjustments, using an Aviv 2000 Circular Dichroism Spectrometer Model 410 (Aviv Biomedical Inc.) (12). CD signal was measured from 250 to 190 nm at 25°C, averaging measurements at every nm for 10 s. Hydrophilins and DMPG were kept on ice until added to samples. Once added, samples were mixed well by mixing and incubated at room temperature for 10 min before spectra were measured. A 1M stock of trehalose and 100 mM stock of SDS were prepared in 10 mM sodium phosphate, pH 7.4 and added to achieve desired concentrations. Proteins were added to samples to achieve a concentration of. 32 mg/ml. For samples with DMPG, DMPG was added to achieve a concentration of 1.32 mM (1.2 mg/ml. Samples were made at a volume of 400 and 300 μL were pipetted into a 0.1 cm cuvette for CD measurements. The cuvette was washed with water and ethanol between measurements, drying completely each time. Samples with protein were blanked against those with the same buffer and concentration of DMPG without hydrophilins. All spectra were converted to units of mean residue molar ellipticity before plotting on graph.

## SGA strain construction

Strains were construction according to published SGA protocol from Tong et al. with minor alterations. In brief, *tps1Δ* was grown overnight as well as every strain in the ATCC deletion collection. Using a 48-density prong, *tps1Δ* was pinned onto solid YEP + Galactose media and the deletion collection were pinned on top. This was allowed to grow overnight at 30°C. Cells were then replica plated onto YEP +Galactose + G418 (100 mg/mL) + Hygromycin (100 mg/mL) to select for diploids. These were grown at 30°C overnight. Diploids were then replica plated onto sporulation media (10 g Potassium Acetate, 1 g Yeast Extract, 0.5 g Galactose, 0.1 g amino acid Histidine, Lysine, Leucine, and Uracil supplement, Zinc Acetate, 0.2 g Raffinose). Cells were sporulated for 5 days. After sporulation, we selected for MATa haploids on SC + Galactose – Histidine – Arginine + Canavanine (50 mg/L) + Thialysine (50 mg/L). Then we selected for G418 resistance on SC + Galactose – Histidine – Arginine + Canavanine (50 mg/L) + Thialysine (50 mg/L) + G418 (100 mg/mL). Finally, we selected for Hygromycin resistance on SC + Galactose – Histidine – Arginine + Canavanine (50 mg/L) + Thialysine (50 mg/L) + G418 (100 mg/mL) + Hygromycin (100 mg/mL).

## High throughput desiccation tolerance assay

Single colonies of newly constructed double mutants are placed into 200 µL of YEP + 2% Galactose in wells of 96-well plates and allowed to grow to saturation at 30°C with agitation. Strains were then pinned using a 48-density prong onto YEP + Galactose plates and allowed to grow for 2 days as non-desiccated controls. 20 µL of each strain was also transferred into new 96-well plates to let air dry for 6 and 30 days at 23°C. Drying was done as previously mentioned. After desiccation, strains are rehydrated in 200 µL of YEP + Galactose and pinned onto solid media as before. Desiccation tolerance is assayed by comparing growth of strains after desiccation with non-desiccated controls.

## Heat tolerance assay

Supplemental Figure 1 (A) Cells (wild type, hsp12Δ, tps1Δ, or tps1Δ hsp12Δ) were grown to midexponential phase (OD <0.5) in non-selective media (YP, 1% yeast extract and 2% peptone) containing 2% glucose at 30C. Cells were plated and grown at either 30 or 37C, or grown for one hour at 34C (pre-heat shock) before growing at either 30 or 37C. (B) Cells (wild type, hsp12Δ, tps1Δ, or tps1Δ hsp12Δ) were grown to midexponential phase (OD <0.5) in non-selective media (YP, 1% yeast extract and 2% peptone) containing 2% glucose at 30C. Cells were then heat-shocked for 30 min at temperatures ranging from 42 to 60C, followed by plating and growing at 30C. Cells were also grown at 34C prior to heat shock.

## 8XΔ strain construction

Knocking out hydrophilin redundancies via cycles of sexual assortment and fluorescence selection (Green Monster Protocol) Adapted from *Suzuki et al. (2011)*. Using deletion collection (a, BY4741), streak out and confirm deletions of interest. *hsp12::G418, gre1::G418, sip18::G418, stf2::G418, ybr016w::G418, yjl144w::G418,* and *ynl190w::G418*. Transform deletions with GFP-URA replacement cassette, plate on SC-URA. Confirm GFP replacements by PCR. Cross deletion GFP-URA strains with (Mat alpha) GMToolkit a and alpha, on a YPD plate, grow overnight. Replica plate on diploid selective media, SC-URA + Clonat (TKalpha) and SC-URA +G418, (TKa). Sporulate diploid strains. Isolate random spores. Recover haploids for each strain in different mating type backgrounds. (a cells should grow on SC-URA-HIS, and alpha cells should grow on SC-URA-LEU.) Sexual cycling of green monsters. Pool deletions Mat a and Mat alpha in 1:1 mix. Take 100 ul from each strain (a and alpha). Mix well. Wash 1x in YPD, resuspend in 1 ml YPD. Spin down at 735 g for 5 min. Incubate mating mix at 30°C for 24 hr. Transfer 100 ul of mating mix into 5 ml of GNA-G418 + Clonat (select for diploids) for 24 hr. Split equally into two tubes. Centrifuge at 735 g for 5 min. To pellet, add 5 ml of SC-HIS or SC-LEU, grow for 24 hr. For GFP induction: Spin down culture and resuspend in 5 ml of SC-HIS or SC-LEU with Doxycycline (10 ug/ml, final) and grow for 2 days at 30°C. Filter 500 ul of GFP induced culture using cells strainer into 1 ml of TE Buffer (pH 7.5) containing Doxycycline (10 ug/ml). Vortex before sorting. Flow Cytometry Cell Sorting. Prepare a BY4741 negative control and the appropiate 1x GFP strain and a 16x GFP control strains. Sort by fluorescent intensity. Confirm by PCR. Strain recovered: hsp12::GFP, gre1::GFP, sip18::GFP, stf2::GFP, ybr016w::GFP, yjl144w::GFP and ynl190w::GFP. *nop6Δ* was deleted from 7XΔ strain with Hygromycin cassette replacement *nop6::Hyg*.

## Acknowledgements

We thank Kevin A Morano, Jeremy Thorner, Jasper Rine and Patrick Gibney for critical reading of our manuscript; members of the Koshland lab for technical support; and Reena Zalpuri at the electron microscope lab at UC Berkeley. We would like to thank Emily J Guinn and Susan Marquee for help with circular dichroism studies. We would also like to thank Fritz Roth at the University of Toronto for the Green Monster Reagents. This work was supported by a grant from the G Harold and Leila Y Mathers Charitable Foundation to SXK and HT, by an NIH grant (GM092813) to DK, and by Damon Runyon Cancer Research Foundation (DRG-2137–12) to GÇ.

## Additional information

### Funding

| Funder | Grant reference number | Author |
|---|---|---|
| G Harold and Leila Y. Mathers Foundation | | Skylar Xantus Kim<br>Hugo Tapia |
| Damon Runyon Cancer Research Foundation | DRG-2137–12 | Gamze Çamdere |
| National Institutes of Health | GM092813 | Douglas Koshland |

The funders had no role in study design, data collection and interpretation, or the decision to submit the work for publication.

### Author contributions

Skylar Xantus Kim, Data curation, Formal analysis, Investigation, Methodology; Gamze Çamdere, Conceptualization, Data curation, Formal analysis, Investigation, Methodology; Xuchen Hu, Formal analysis, Investigation; Douglas Koshland, Conceptualization, Data curation, Formal analysis, Supervision, Funding acquisition, Validation, Writing—original draft, Writing—review and editing; Hugo Tapia, Conceptualization, Data curation, Formal analysis, Supervision, Funding acquisition, Validation, Investigation, Visualization, Methodology, Writing—original draft, Project administration, Writing—review and editing

### Author ORCIDs

Douglas Koshland  https://orcid.org/0000-0003-3742-6294
Hugo Tapia  http://orcid.org/0000-0003-1901-2151

### Decision letter and Author response

Decision letter https://doi.org/10.7554/eLife.38337.023
Author response https://doi.org/10.7554/eLife.38337.024

## Additional files

### Supplementary files

• Supplementary file 1. Hydrophilin protein purifaction. Hsp12 and Stf2 are bacterial derived protein preps. Gel stained with Coomassie Blue demonstrates purity of protein preps.
DOI: https://doi.org/10.7554/eLife.38337.018

• Supplementary file 2. Synthetic Genetic Array Desiccation Screen – tps1Δ. (A) Three different groups where identified from our SGA - tps1Δ screen. (1) gene deletions that were synthetic lethal with tps1Δ: failed to grow completely. (2) gene deletions that lead to desiccation sensitivity with tps1Δ, and (3) gene deletions that allowed tps1Δ to grow after 30 days of drying: suppressors. List of genes attached as 1–3. (B) Breakdown of desiccation sensitive tps1Δ double mutants into categories based on cellular function. Each desiccation sensitive double mutant was placed into a category based on their cellular function (GO).
DOI: https://doi.org/10.7554/eLife.38337.019

• Supplementary file 3. Strain Table. Strains used in this study.
DOI: https://doi.org/10.7554/eLife.38337.020

• Transparent reporting form
DOI: https://doi.org/10.7554/eLife.38337.021

### Data availability

All data generated or analysed during this study are included in the manuscript and supporting files.

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
