## [Decision Letter]

[Editors’ note: a previous version of this study was rejected after peer review, but the authors submitted for reconsideration. The first decision letter after peer review is shown below.]

Thank you for submitting your work entitled "Synergism between a sugar and a small intrinsically disordered protein mitigate the lethal stresses of severe water loss" for consideration by *eLife*. Your article has been reviewed by two peer reviewers, and the evaluation has been overseen by a Reviewing Editor and a Senior Editor. The reviewers have opted to remain anonymous.

Our decision has been reached after consultation between the reviewers. Based on these discussions and the individual reviews below, we regret to inform you that your work will not be considered further for publication in *eLife*.

Although the reviewers felt that the area was of interest, the felt that as it stands, the manuscript lacked sufficient depth to be considered for publication in *eLife*. The two reviewers are experts in the field of desiccation tolerance and as you can see, they both feel that there is a considerable literature out there, and that moving forward, it would be good to put your experiments in the context of these experiments.

*Reviewer #1:*

This manuscript by Kim et al. investigates the role of trehalose and the hydrophilin Hsp12 in desiccated budding yeast. The authors show that these two molecules prevent the aggregation and stabilize the activity of model proteins. They also propose a role for Hsp12 as a remodeler of membranes. There are several interesting observations in this paper. However, there are also many problems that significantly reduce my enthusiasm for the paper. The biggest problem is that similar findings have already been reported previously in budding yeast but also in other model organisms. I am therefore not sure whether this manuscript generates the level of advance that I would associate with a paper in *eLife* paper. To be truly novel, the paper would have to provide some deeper insight into the molecular mechanisms underlying the synergistic protective effect of trehalose and Hsp12.

1) Figure 2A. The authors conclude from these data that luciferase is denatured. However, the amount of luminescence is generally quite low. Therefore, is it possible that water is needed for enzyme activity? Can the authors exclude this possibility? If they cannot, I suggest that they show directly that luciferase is denatured, for example by using dyes that bind to exposed hydrophobic regions.

2) I think that yeast prions are not a good model for looking at how desiccation affects protein aggregation. The reason is that, in their yeast strains, the prion proteins are already in an aggregated state. Therefore, the authors can only look at propagation and not at de novo formation of prions. So what is the authors explanation for how trehalose affects [*PSI*^+^] prions? They conclude: "These results suggest that trehalose is more effective than Hsp12 in preventing aggregation of this cytoplasmic prion." How can the authors conclude from these data that trehalose prevents aggregation? There are fewer prions in the absence of trehalose. This suggests that trehalose promotes aggregation and/or prion propagation. One possibility is that trehalose protects the [*PSI*^+^] aggregates from damage and that the prion state then gets lost from the cells in the absence of trehalose after recovery from desiccation. I think that this is very unlikely, because these aggregates are highly stable and probably not affected much by water loss. The presented data are also consistent with another scenario. The data in Figure 2C could also be explained by a synthetic lethality effect in which the [*PSI*^+^] cells are more sick in the absence of trehalose (did the authors correct for cell numbers or did they only count the number of cells on adenine-deficient plates?). The same reasoning applies to the [*GAR*^+^] data.

3) Why did the authors perform the in vitro assay with citrate synthase and the in vivo assay with luciferase? Luciferase misfolding and aggregation is a well-established in vitro assay for protein misfolding and aggregation. Can the authors perform an in vitro assay with luciferase to make the in vitro and in vivo data more comparable?

4) Is there a synergistic effect of trehalose and Hsp12 in the in vitro citrate synthase assay as found for luciferase expressed in yeast?

5) What is the significance of these fragmented liposomes in Figure 3B? To me it looks as if Hsp12 does not protect lipid membranes but destroys them. How do yeast membranes look in the absence/presence of Hsp12 during and after desiccation?

6) Figure 4C. These data are of poor quality. Can the authors quantify the amount of lipids? Also, can they demonstrate statistical significance? The same is true for Figure 3A.

*Reviewer #2:*

This manuscript is dedicated to investigation of the molecular basis of desiccation tolerance in *Saccharomyces cerevisiae*. Previously, the authors have shown that a disaccharide trehalose is necessary for the long-term desiccation tolerance but is not required for the short one. On the other hand, a previous screen for non-essential genes that are required for the short-term tolerance was not successful. Thus, the authors decided to perform a "synthetically lethal" screen. They used a strain that cannot produce trehalose and introduced into it deletions of about 5000 non-essential genes. These double mutants were investigated for their ability to survive short and long desiccation while being in the stationary phase. The screen has revealed three classes of mutants: 1) Inviable when subjected to short-term desiccation; 2) Inviable without application of stress; 3) Suppressors of trehalose deficient mutant subjected to long-term desiccation. This study describes the role of Hsp12, a gene from class I, in the desiccation tolerance. It is shown that trehalose and Hsp12p, a member of the hydrophilin family of proteins, are acting synergistically. Proteostatic properties of the protein and its possible interaction with membranes are investigated. In addition, the authors provide data that another member of hydrophilins, Stf2, does not play any role in the desiccation tolerance.

I think that the performed screen is potentially very useful for understanding molecular mechanisms of the desiccation tolerance in the future. I have, however, several serious concerns on validity of some conclusions drawn and on the overall novelty of the findings presented.

1) In fact, Hsp12p is a homolog of class III LEA (Late embryonic abundant) proteins in plants. There is a vast amount of literature on importance of these proteins in the draught resistance. Also in *C. elegans*, IDPs (Intrinsically disordered proteins) play a role in desiccation tolerance. In particular, RNAi against lea-1 and dur-1 (two members of LEA family) leads to very strong sensitivity to desiccation (Erkut at al., 2013). Moreover, there are several reports on importance of hydrophilins/LEAs for desiccation tolerance in the budding yeast (Lopez-Martinez et al., Yeast, 30, 319; Lopez-Martinez et al., 2012). Thus, the requirement of these proteins for the tolerance is well established. What is not clear, however, is the molecular function of LEAs in the living organism. Involvement in proteostasis, disaggregation of proteins, interaction with membranes and many other functions (including recently the phase separation) are proposed. Again, huge number of papers are dealing with the topic. Unfortunately, this manuscript, does not provide any additional, unambiguous information on the molecular mechanism of Hsp12p action.

2) The interaction between *tps1Δ* and *hsp12Δ* is described in purely genetic frame. Are trehalose and Hsp12p connected also biochemically? Does amount of trehalose depend on the protein? In fact, the quantitative information on amounts of Hsp12p before and during the desiccation is totally missing. Is it induced during the desiccation? If yes, on which level? Interestingly, LEA-1 in *C. elegans* is the major protein that is elevated during the preconditioning of the worm for severe desiccation (Erkut at al., 2013). The authors say that the expression of Hsp12p is low. This would question the proposed non-enzymatic role of the protein in proteostasis or membrane remodeling.

3) The authors show that Hsp12p when incubated with liposomes, induces their vesiculation into nanovesicles. Or in other words, their disintegration. Does this phenomenon has anything to do with the desiccation tolerance? Without showing this process in the cell, this property of the protein remains just an interesting observation.

4) It is shown in the Figure 4 that another hydrophilin, Stf2p, is not required for the desiccation tolerance. This is in a direct contrast with a previously published paper (Lopez-Martinez et al., 2012). The authors should at least comment on this controversy.

[Editors’ note: what now follows is the decision letter after the authors submitted for further consideration.]

Thank you for submitting your article "Synergy between the small intrinsically disordered protein Hsp12 and trehalose sustain viability after desiccation" for consideration by *eLife*.

The two reviewers feel that the manuscript is significantly improved, and together with the reviewing editor, would like to publish your paper, assuming you can make the necessary revisions below.

1) It is still unclear whether Hsp12 has a membrane remodeling function in vivo or not. All they provide are in vitro experiments, the functional significance of which is unclear. They have to change the text to make this clear. For example, in the Abstract: "We also identify a novel role for Hsp12 as a membrane remodeler, a protective feature not shared by another yeast hydrophilin, suggesting that sIDPs have distinct biological functions." They do not identify a function as membrane modeler in cells and they do not show whether this particular in vitro activity of Hsp12 is protective in desiccated cells. For this, they need a mutant that lacks the ability to remodel membranes that they then test in cells. Since they have not done this experiment, they have to tone down their conclusions regarding the membrane remodeling function.

2) Section headline: "Hsp12 and trehalose can act synergistically in membrane proteostasis." It is unclear whether the observed effect on prion propagation is because of altered membrane proteostasis. Therefore, they have to remove any claims that the prion assay reports on membrane proteostasis or proteostasis in general.

3) "Without trehalose, the [*PSI*^+^] prion hyper-aggregates in mother cells suggesting that trehalose functions during desiccation as an anti-protein aggregation factor as suggested by other in vitro and in vivo studies." The authors provide no evidence for prion hyper-aggregation in mother cells. They have to remove this claim. Also, what other in vitro and in vivo studies are they referring to here?

4) Is the phenotype of *HSP12* based on the vesiculation of membranes? I think the authors either should provide EM images of desiccated yeast (with and without *HSP12*) or take out from the Abstract the claim that they provide a novel role of *HSP12* as a membrane remodeler. Very short hydration and then cryofixation with substitution could be a method. Here is the reference to a similar approach (Giddings et al., 2001, Methods Cell Biol.).

---

## [Author Response]

[Editors’ note: the author responses to the first round of peer review follow.]

Reviewer #1:This manuscript by Kim et al. investigates the role of trehalose and the hydrophilin Hsp12 in desiccated budding yeast. The authors show that these two molecules prevent the aggregation and stabilize the activity of model proteins. They also propose a role for Hsp12 as a remodeler of membranes. There are several interesting observations in this paper. However, there are also many problems that significantly reduce my enthusiasm for the paper. The biggest problem is that similar findings have already been reported previously in budding yeast but also in other model organisms. I am therefore not sure whether this manuscript generates the level of advance that I would associate with a paper in eLife paper. To be truly novel, the paper would have to provide some deeper insight into the molecular mechanisms underlying the synergistic protective effect of trehalose and Hsp12.1) Figure 2A. The authors conclude from these data that luciferase is denatured. However, the amount of luminescence is generally quite low. Therefore, is it possible that water is needed for enzyme activity? Can the authors exclude this possibility? If they cannot, I suggest that they show directly that luciferase is denatured, for example by using dyes that bind to exposed hydrophobic regions.

We agree with reviewer #1. Water is indeed required for enzymatic activity. The luminescence measured in Figure 2A (now Figure 2—figure supplement 1A) was measured after rehydration (in the presence of water) of desiccated cells. What the assay measures is the ability of either trehalose or Hsp12 to protect luciferase activity in the absence of water measured via luminescence after water is added back.

2) I think that yeast prions are not a good model for looking at how desiccation affects protein aggregation. The reason is that, in their yeast strains, the prion proteins are already in an aggregated state. Therefore, the authors can only look at propagation and not at de novo formation of prions. So what is the authors explanation for how trehalose affects [PSI^+^] prions? They conclude: "These results suggest that trehalose is more effective than Hsp12 in preventing aggregation of this cytoplasmic prion." How can the authors conclude from these data that trehalose prevents aggregation? There are fewer prions in the absence of trehalose. This suggests that trehalose promotes aggregation and/or prion propagation. One possibility is that trehalose protects the [PSI^+^] aggregates from damage and that the prion state then gets lost from the cells in the absence of trehalose after recovery from desiccation. I think that this is very unlikely, because these aggregates are highly stable and probably not affected much by water loss. The presented data are also consistent with another scenario. The data in Figure 2C could also be explained by a synthetic lethality effect in which the [PSI^+^] cells are more sick in the absence of trehalose (did the authors correct for cell numbers or did they only count the number of cells on adenine-deficient plates?). The same reasoning applies to the [GAR^+^] data.

We have previously established yeast prions as a model for looking at how desiccation affects protein aggregation. (Tapia and Koshland, 2014). Indeed, other independent groups have recently used our desiccation prion propagation assay successfully to assess the propagation of wild type yeast prions. (Ramakrishnan et al., 2016, Frontiers in Ecology and Evolution).

3) Why did the authors perform the in vitro assay with citrate synthase and the in vivo assay with luciferase? Luciferase misfolding and aggregation is a well-established in vitro assay for protein misfolding and aggregation. Can the authors perform an in vitro assay with luciferase to make the in vitro and in vivo data more comparable?

The assay was done with different model substrates to demonstrate the versatility of protection by our desiccation effectors. We agree that in vitroluminescence assays would complement our data set; unfortunately, luciferase was extremely sensitive to in vitrodenaturation via desiccation and neitherluminescence nor aggregation were recovered to measurable levels after drying alone or in the presence of any protectant.

4) Is there a synergistic effect of trehalose and Hsp12 in the in vitro citrate synthase assay as found for luciferase expressed in yeast?

Figure 2—figure supplement 1C-D. They synergize to protect activity but not aggregation, yet the recovered activity is the same as we see with Hsp12 alone.

5) What is the significance of these fragmented liposomes in Figure 3B? To me it looks as if Hsp12 does not protect lipid membranes but destroys them. How do yeast membranes look in the absence/presence of Hsp12 during and after desiccation?

Despite looking like destruction of liposomes, we suggest that the remodeling activity of Hsp12 is limited to damaged membranes. The observed in vitro vesiculation activity is likely controlled via either localization or concentration. We have added a model of possible function of Hsp12 to the Discussion.

Yeast membranes are disturbed by the absence of Hsp12 as can be seen in published work upon heat shock. (Welker et al., 2010, Molecular Cell).

6) Figure 4C. These data are of poor quality. Can the authors quantify the amount of lipids? Also, can they demonstrate statistical significance? The same is true for Figure 3A.

Both Figures 3A and 4C (now Figures 4A, B and Figure 5D) are now of much improved quality.

Reviewer #2:[…] I think that the performed screen is potentially very useful for understanding molecular mechanisms of the desiccation tolerance in the future. I have, however, several serious concerns on validity of some conclusions drawn and on the overall novelty of the findings presented.1) In fact, Hsp12p is a homolog of class III LEA (Late embryonic abundant) proteins in plants. There is a vast amount of literature on importance of these proteins in the draught resistance. Also in C. elegans, IDPs (Intrinsically disordered proteins) play a role in desiccation tolerance. In particular, RNAi against lea-1 and dur-1 (two members of LEA family) leads to very strong sensitivity to desiccation (Erkut at al., 2013). Moreover, there are several reports on importance of hydrophilins/LEAs for desiccation tolerance in the budding yeast (Lopez-Martinez et al., Yeast, 30, 319; Lopez-Martinez et al., 2012). Thus, the requirement of these proteins for the tolerance is well established. What is not clear, however, what is the molecular function of LEAs in the living organism. Involvement in proteostasis, disaggregation of proteins, interaction with membranes and many other functions (including recently the phase separation) are proposed. Again, huge number of papers are dealing with the topic. Unfortunately, this manuscript, does not provide any additional, unambiguous information on the molecular mechanism of Hsp12p action.

The Introduction and Discussion have been changed to address the above points as well as to differentiate the novelty of our study.

2) The interaction between tps1Δ and hsp12Δ is described in purely genetic frame. Are trehalose and Hsp12p connected also biochemically? Does amount of trehalose depend on the protein? In fact, the quantitative information on amounts of Hsp12p before and during the desiccation is totally missing. Is it induced during the desiccation? If yes, on which level? Interestingly, LEA-1 in C. elegans is the major protein that is elevated during the preconditioning of the worm for severe desiccation (Erkut at al., 2013). The authors say that the expression of Hsp12p is low. This would question the proposed non-enzymatic role of the protein in proteostasis or membrane remodeling.

Both trehalose and Hsp12 are regulated by the same pathway (Environmental Stress Response). We have now added protein blots demonstrating the amounts of hydrophilins (Hsp12, Stf2) in cells prior to desiccation that induce desiccation tolerance (Figure 4B). Neither Hsp12 nor trehalose are induced by the actual act of drying. In the case of wild type desiccation tolerance, yeast cells must first cells must first be in a quiescent state to be tolerant. This is shared by many anhydrobiotes; the nematode must be pre-conditioned ‘dauer’ state and the tardigrade in its pre-conditioned tun state. Expression of Hsp12 is not detected in logarithmically growing cells but induced to extremely high levels during ‘quiescence/starvation’ the preconditioning stage of yeast.

3) The authors show that Hsp12p when incubated with liposomes, induces their vesiculation into nanovesicles. Or in other words, their disintegration. Does this phenomenon has anything to do with the desiccation tolerance? Without showing this process in the cell, this property of the protein remains just an interesting observation.

It has proved difficult to visualize desiccated cells via microscopy (technical limitations). The vesiculation activity observed is so potent that we believe it to be an essential role of Hsp12 function. Given the phenotype of Hsp12 to confer desiccation tolerance and its synergism with trehalose, vesiculation via Hsp12 will likely play a role in desiccation tolerance.

4) It is shown in the Figure 4 that another hydrophilin, Stf2p, is not required for the desiccation tolerance. This is in a direct contrast with a previously published paper (Lopez-Martinez et al., 2012). The authors should at least comment on this controversy.

Desiccation tolerance assays are not quite comparable. Lopez-Martinez et al. rehydrate their dehydrated cells in 37°C media, adding another level of insult (heat shock) to their assay. This information has been added to the manuscript. Maybe Stf2 function is required under desiccated/heat shock conditions but not desiccation alone. Also, Lopez-Martinez et al. desiccate their yeast cells for only approx. 20 hours by air drying, an amount of time insufficient for cells to reach desiccation. We allow our cells a minimum of two days in the dry state via air drying to achieve full desiccation (less than two days of drying only leads to small drops in viability and intracellular content remains higher than 5%). Lopez-Martinez et al. are likely examining dehydrated and not desiccated cells.

[Editors' note: the author responses to the re-review follow.]

The two reviewers feel that the manuscript is significantly improved, and together with the reviewing editor, would like to publish your paper, assuming you can make the necessary revisions below.1) It is still unclear whether Hsp12 has a membrane remodeling function in vivo or not. All they provide are in vitro experiments, the functional significance of which is unclear. They have to change the text to make this clear. For example, in the Abstract: "We also identify a novel role for Hsp12 as a membrane remodeler, a protective feature not shared by another yeast hydrophilin, suggesting that sIDPs have distinct biological functions." They do not identify a function as membrane modeler in cells and they do not show whether this particular in vitro activity of Hsp12 is protective in desiccated cells. For this, they need a mutant that lacks the ability to remodel membranes that they then test in cells. Since they have not done this experiment, they have to tone down their conclusions regarding the membrane remodeling function.

We have fixed our text in our paper and toned down the results. We rewrite the observation as only an in vitro observation at the time.

2) Section headline: "Hsp12 and trehalose can act synergistically in membrane proteostasis." It is unclear whether the observed effect on prion propagation is because of altered membrane proteostasis. Therefore, they have to remove any claims that the prion assay reports on membrane proteostasis or proteostasis in general.

We have fixed the text. Instead of membrane proteostasis, it is rewritten as “Hsp12 and trehalose act synergistically to propagate a membrane prion”.

3) "Without trehalose, the [PSI^+^] prion hyper-aggregates in mother cells suggesting that trehalose functions during desiccation as an anti-protein aggregation factor as suggested by other in vitro and in vivo studies." The authors provide no evidence for prion hyper-aggregation in mother cells. They have to remove this claim. Also, what other in vitro and in vivo studies are they referring to here?

The sentence has been removed from text.

4) Is the phenotype of HSP12 based on the vesiculation of membranes? I think the authors either should provide EM images of desiccated yeast (with and without HSP12) or take out from the Abstract the claim that they provide a novel role of HSP12 as a membrane remodeler. Very short hydration and then cryofixation with substitution could be a method. Here is the reference to a similar approach (Giddings et al., 2001, Methods Cell Biol.).

We have fixed our text in our paper and toned down the results. We rewrite the observation as only an in vitro observation at the time.